# Severe Hypertriglyceridaemia and Chylomicronaemia Syndrome—Causes, Clinical Presentation, and Therapeutic Options

**DOI:** 10.3390/metabo13050621

**Published:** 2023-04-30

**Authors:** Bilal Bashir, Jan H. Ho, Paul Downie, Paul Hamilton, Gordon Ferns, Dev Datta, Jaimini Cegla, Anthony S. Wierzbicki, Charlotte Dawson, Fiona Jenkinson, Hannah Delaney, Michael Mansfield, Yee Teoh, Zosia Miedzybrodzka, Haya Haso, Paul N. Durrington, Handrean Soran

**Affiliations:** 1Faculty of Biology Medicine and Health, University of Manchester, Manchester M13 9PL, UK; 2Department of Endocrinology, Diabetes & Metabolism, Manchester University NHS Foundation Trust, Manchester M13 9WL, UK; 3Department of Endocrinology, The Christie NHS Foundation Trust, Manchester M20 4BX, UK; 4Department of Laboratory Medicine, Salisbury NHS Foundation Trust, Salisbury SP2 8BJ, UK; 5Centre for Medical Education, Queen’s University Belfast, Belfast BT7 1NN, UK; 6Department of Clinical Biochemistry, Belfast Health and Social Care Trust, Belfast BT13 1FD, UK; 7Brighton and Sussex Medical School, Brighton BN1 9PH, UK; 8Lipid Unit, University Hospital Llandough, Cardiff CF64 2XX, UK; 9Division of Diabetes, Endocrinology and Metabolism, Imperial College London, London SW7 2BX, UK; 10Department of Metabolic Medicine and Chemical Pathology, Guy’s and St. Thomas’ Hospitals, London SE1 7EH, UK; 11Department of Metabolic Medicine, Queen Elizabeth Hospital NHS Foundation Trust, Birmingham PE30 4ET, UK; 12Clinical Biochemistry and Metabolic Medicine, Royal Victoria Infirmary, Newcastle upon Tyne NE1 4LP, UK; 13Department of Clinical Chemistry, Sheffield Teaching Hospitals NHS Foundation Trust, Sheffield S10 2JF, UK; 14Leeds Centre for Diabetes & Endocrinology, Leeds Teaching Hospitals NHS Trust, Leeds LS9 7TF, UK; 15Department of Chemical Pathology & Metabolic Medicine, Wrexham Maelor Hospital, Wrexham LL13 7TD, UK; 16Department of Medical Genetics, School of Medicine, Medical Sciences and Nutrition, University of Aberdeen, Aberdeen AB24 3FX, UK; 17School of Medicine, University of Kurdistan Hewler, Erbil 44001, Iraq

**Keywords:** hypertriglyceridaemia, chylomicronaemia syndrome, pancreatitis, atherosclerosis, microvascular complications, volanesorsen

## Abstract

We have reviewed the genetic basis of chylomicronaemia, the difference between monogenic and polygenic hypertriglyceridaemia, its effects on pancreatic, cardiovascular, and microvascular complications, and current and potential future pharmacotherapies. Severe hypertriglyceridaemia (TG > 10 mmol/L or 1000 mg/dL) is rare with a prevalence of <1%. It has a complex genetic basis. In some individuals, the inheritance of a single rare variant with a large effect size leads to severe hypertriglyceridaemia and fasting chylomicronaemia of monogenic origin, termed as familial chylomicronaemia syndrome (FCS). Alternatively, the accumulation of multiple low-effect variants causes polygenic hypertriglyceridaemia, which increases the tendency to develop fasting chylomicronaemia in presence of acquired factors, termed as multifactorial chylomicronaemia syndrome (MCS). FCS is an autosomal recessive disease characterized by a pathogenic variant of the *lipoprotein lipase* (*LPL*) gene or one of its regulators. The risk of pancreatic complications and associated morbidity and mortality are higher in FCS than in MCS. FCS has a more favourable cardiometabolic profile and a low prevalence of atherosclerotic cardiovascular disease (ASCVD) compared to MCS. The cornerstone of the management of severe hypertriglyceridaemia is a very-low-fat diet. FCS does not respond to traditional lipid-lowering therapies. Several novel pharmacotherapeutic agents are in various phases of development. Data on the correlation between genotype and phenotype in FCS are scarce. Further research to investigate the impact of individual gene variants on the natural history of the disease, and its link with ASCVD, microvascular disease, and acute or recurrent pancreatitis, is warranted. Volanesorsen reduces triglyceride concentration and frequency of pancreatitis effectively in patients with FCS and MCS. Several other therapeutic agents are in development. Understanding the natural history of FCS and MCS is necessary to rationalise healthcare resources and decide when to deploy these high-cost low-volume therapeutic agents.

## 1. Introduction

Triglycerides (TGs) are the main constituents of dietary fat and form the main source of energy stored in the human body. Diet is the major exogenous source of TG, whilst the liver is the principal endogenous source. Dietary TGs are broken down in the upper gastrointestinal tract into glycerol and free fatty acids, taken into enterocytes, reformed into TGs, and assembled into chylomicrons. Following packaging into chylomicrons and entry into the systemic circulation, chylomicrons are progressively hydrolysed in capillary beds by the action of lipoprotein lipase (LPL) facilitated by its regulatory proteins. Continuous hydrolysis of TG leads to a reduction in the size of chylomicron particles. Chylomicron remnant (CR) particles undergo hepatic clearance, primarily involving low-density lipoprotein (LDL) receptors, LDL receptor-related protein-1 (LRP-1), and heparan sulphate proteoglycans (syndecan 1) (Figure 1a). Cholesterol and TGs acquired by the liver are assembled and secreted into the systemic circulation in the form of triglyceride-rich lipoproteins (TRL), including very-low-density lipoproteins (VLDL). The metabolic kinetics of VLDL is similar to that of chylomicron particles (Figure 1a,b) [1]. 

An increase in circulating TGs is a normal physiological response after a meal. Ingested fat is subjected to the above-mentioned catabolic pathway, leading to the clearance of TRL from systemic circulation. However, in certain conditions, TGs are not cleared either due to suboptimal enzyme activity, hepatic overproduction, or reduced hepatic clearance that leads to pathological hypertriglyceridaemia. Healthy TG levels are consistently defined as <1.7 mmol/L; however, the severity of hypertriglyceridaemia is less well-defined (Table 1) [2,3,4,5]. 

**Table 1 metabolites-13-00621-t001:** Definitions of hypertriglyceridaemia.

2018 AHA/ACC Clinical Practice Guidelines *	NCEP ATP III	Endocrine Society	ESC/EAS Guidelines
Normal: <2.0 mmol/L Moderate: 2.0–5.6 mmol/L Severe: >5.6 mmol/L	Normal: <1.7 mmol/L Borderline high: 1.7–2.3 mmol/L High: 2.3–5.6 mmol/L Very High: >5.6 mmol/L	Normal: <1.7 mmol/L Mild: 1.7–2.3 mmol/L Moderate: 2.3–11.2 mmol/L Severe: 11.2–22.4 mmol/L Very severe: >22.4 mmol/L	Normal: <1.7 mmol/L Mild to Moderate: >1.7 mmol/L Severe: >10.0 mmol/L

* In the 2021 American College of Cardiology (ACC) Expert Consensus Decision Pathway on the Management of ASCVD Risk Reduction in Patients with Persistent Hypertriglyceridemia, persistent hypertriglyceridaemia was defined as a fasting TG level of >1.7mmol/L following a minimum of 4 to 12 weeks of lifestyle intervention, a stable dose of maximally tolerated statin when indicated, as well as evaluation and management of secondary causes of hypertriglyceridaemia [6]. ACC: American College of Cardiology; AHA: American heart association; EAS: European atherosclerosis society; ESC: European society of cardiology; NCEP ATP III: National cholesterol education programme Adult Treatment plan III.

**Figure 1 metabolites-13-00621-f001:**
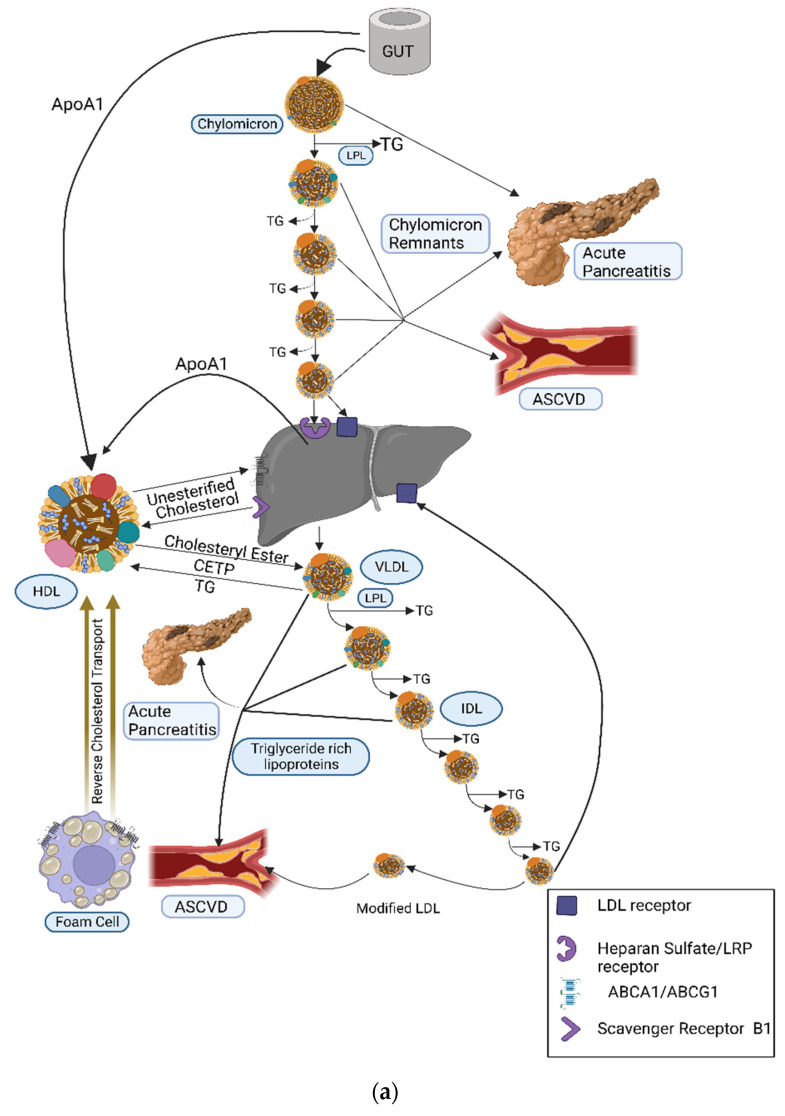
(**a**) Dietary TGs and cholesterol are hydrolysed by lingual, gastric, pancreatic, and intestinal lipases into free fatty acids (FFA) and glycerol that are transported from the intestinal lumen to enterocytes where they are re-esterified and packaged into the transport cargo, chylomicrons. These enter circulation via the lymphatic system (the thoracic duct) and, thence, to the subclavian vein. In the systemic circulation, TG in chylomicrons is hydrolysed by lipoprotein lipase (LPL) which is expressed in high concentrations on the capillary surface of muscle and adipose tissue. Continuous hydrolysis of TG in chylomicron particles reduces its size and transfers apolipoproteins from the chylomicron surface to other lipoproteins while retaining ApoB48 and ApoE. Remnant particles undergo hepatic clearance, primarily involving LDL receptors, LDL receptor-related protein-1 (LRP-1), and heparan sulphate proteoglycans (syndecan 1). ApoE is important for this receptor-mediated uptake and clearance of chylomicron remnants, the failure of which leads to diminished chylomicron clearance and resultant hypertriglyceridaemia (as seen in dysbetalipoproteinaemia). CR and other TRL can penetrate the vascular endothelium and, unlike LDL, can be taken up by macrophages directly without chemical modification, hence contributing to atherogenesis. CETP facilitates the transfer of CE from HDL to TRL and TG from TRL to HDL. LDL represents major circulatory cholesterol and is delivered to most of the tissues and liver via the LDL receptor. Excess cholesterol is removed from cells via HDL and is transferred either to other lipoprotein subfractions or directly to the liver. Adapted from Bhatnagar D, Soran H, Durrington PN. Hypercholesterolaemia and its management [7]. (**b**) Dietary fat is absorbed from the intestine and assembled in chylomicrons. Hydrolysis of TG by LPL expressed on the capillary surface of fat and muscle cells leads to the production of CR that are taken up by the liver. The liver produces TRL that undergoes hydrolysis by LPL, leading to a progressive reduction in its size with loss of TG from its core, leading to the production of IDL and LDL. LMF1is a membrane-spanning protein that is responsible for the maturation, stabilisation, and transport of LPL to the capillary endothelial surface. GPIHBP1 binds LPL from the subendothelial interstitial space, transporting and anchoring it to the luminal surface of vascular endothelial cells. ApoC2 acquired from HDL acts as an activator of LPL and is important in regulating LPL activity and chylomicron catabolism. ApoA5 acts as an activator of LPL and the loss of function of ApoA5 leads to reduced LPL activity. ApoC3 inhibits LPL activity and apoE-mediated hepatic uptake of chylomicron and VLDL remnants. ANGPLT3 inhibits LPL activity. ABCA1: ATP-binding cassette transporter protein 1; ABCG1: ATP-binding cassette transporter member 1 of subfamily G; ANGPTL3: angiopoietin-like 3; Apo: apolipoprotein; CE: cholesteryl ester; CETP: cholesteryl ester transfer protein; CR: chylomicron remnants; FFA: free fatty acids; GPIHBP1: glycosylphosphatidylinositol-anchored HDL-binding protein1; HDL, high-density lipoprotein; IDL: intermediate-density lipoprotein; LDL: low-density lipoprotein; LPL: lipoprotein lipase; LMF1: lipase maturation factor 1; LRP, lipoprotein-related receptor; TG: triglycerides; TRL: triglyceride-rich lipoproteins; VLDL: very-low-density lipoprotein; VLDL, very-low-density lipoprotein.

Conventionally, TG levels > 10 mmol/L are often regarded as severe hypertriglyceridaemia (SHTG). In centres that have not adapted to SI units of measurements, a TG level > 1000 mg/dL (11.3 mmol/L) is often regarded as SHTG. SHTG is often accompanied by intermittent or persistent chylomicronaemia. Fasting chylomicronaemia may occur due to pathological variation in lipoprotein lipase (LPL) or genes that govern the activity of LPL. This monogenic form of chylomicronaemia is often accompanied by a constellation of symptoms and is termed as familial chylomicronaemia syndrome (FCS). Alternatively, SHTG may be polygenic, i.e., when compounded by secondary factors, it aggravates the severity of hypertriglyceridaemia. This is termed as multifactorial chylomicronaemia syndrome (MCS).

In this review, we aimed to review the aetiology of SHTG and chylomicronaemia, its genetic basis, differences between FCS and MCS, the effect of SHTG on pancreatic, cardiovascular, and microvascular complications, current and potential future pharmacotherapies.

## 2. Epidemiology of Severe Hypertriglyceridaemia

(Mild hypertriglyceridaemia is common among adults; however, SHTG, which is often defined by TG levels >10 mmol/L or 1000 mg/dL, is rare. Although the prevalence of hypertriglyceridaemia in the American, European, and Russian population is 25–30%, the prevalence of SHTG is <1% [8,9,10,11]. Using linked electronic health records between primary care data, hospital admissions, and death registry data, the prevalence of SHTG was reported to be 0.21% in England [12]. Similarly, data extracted from Norwegian surveys (*n =* 619,990) and a large laboratory database from France (*n* = 100,322) suggest that the prevalence of SHTGis 0.1% [13,14]. Hypertriglyceridaemia can be primary, having a strong genetic basis, or acquired, in which the concomitant presence of secondary factors or drugs renders an individual more susceptible to developing hypertriglyceridaemia. Almost all cases of SHTG developing in the early years of life are monogenic, however mild, and moderate hypertriglyceridaemia is usually polygenic [15]. 

## 3. Primary Hypertriglyceridaemia

Primary hypertriglyceridaemia has a complex genetic basis. It may or may not be a result of a single gene variant and has an intricate genotype–phenotype correlation. FCS is a rare autosomal recessive disorder where SHTG occurs due to homozygous, compound heterozygous, or double heterozygous pathogenic variants in *LPL, apolipoprotein C2 (ApoC2), apolipoproteinA5 (ApoA5), lipase maturation factor 1 (LMF1), glycosylphosphatidylinositol-anchored HDL-binding protein 1 (GPIHBP1), glycerol-3-phosphate dehydrogenase-1* (*GPD1), or cyclic AMP-responsive element-binding protein 3-like protein 3* (*CREB3L3)*. However, carriers of heterozygous disease-causing variants exhibit a wide range of phenotypes, ranging from normal TG to SHTG due to variable co-inheritance of different low-effect hypertriglyceridaemia-causing variants [15]. There can be large inter-individual variability in TG concentrations between carriers of the same variants even within the same family [15], suggesting either an association with other gene loci or the influence of acquired factors. Most of these genetic variants are common in the population, with a small effect size of increase in TG levels up to 0.5 mmol/L per allele, although larger effect sizes of up to 2.3 mmol/L per allele have also been reported in some rare variants [16]. The severity of hypertriglyceridaemia is, therefore, dependent on the balance between the common genetic variants with small effect and the rare genetic variants with larger effect size. For example, a single extremely rare variant with a large effect size can lead to SHTG, e.g., FCS, whilst common variants with a small effect size can also cumulatively increase triglyceride levels, and both can be compounded by acquired factors such as diet, alcohol, obesity, insulin resistance, or medications [15]. 

### 3.1. Chylomicronaemia Syndromes

In normal physiology, chylomicrons are cleared from circulation 3–4 h post-prandially. The persistence of chylomicron particles whilst fasting (12–14 h) may be associated with SHTG [17]. Chylomicronaemia syndrome (CS) is defined as the presence of one of the following clinical abnormalities along with chylomicronaemia and SHTG [18]:Eruptive xanthomata;Lipemia retinalis;Recurrent abdominal pain;Acute/chronic pancreatitis;Hepatosplenomegaly;Neuropsychiatric and cognitive complications.

CS can be familial or multifactorial. FCS is a rare monogenic disorder. The exact prevalence of FCS is subject to debate with an estimated prevalence of 1:100,000–1,000,000 [19], though the exact prevalence may be high due to underdiagnosis and limited availability of genetic test resources. It is characterised by a marked reduction in LPL activity. Affected patients typically present first in childhood, or early adulthood, with SHTG, recurrent episodes of acute pancreatitis (AP), and resistance to lipid-lowering treatment [18]. Most cases of chylomicronaemia, however, are polygenic, secondary to the clustering of multiple genetic variants, with the cumulative effect of resulting in the tendency to develop chylomicronaemia and hypertriglyceridaemia in the presence of secondary factors. This is termed multifactorial chylomicronaemia syndrome (MCS) [20]. In contrast to FCS, MCS has a prevalence of 1:600, has a milder phenotype, a relatively lower risk of AP at equivalent TG concentrations, and is generally more responsive to diet, the removal of secondary factors, and lipid-lowering treatment [20]. Regardless of aetiology, the most serious clinical manifestation of CS is AP. The prevalence of AP in chylomicronaemia and SHTG (>11.3 mmol/L) is 5%, which rises to 15% in very severe hypertriglyceridaemia (22.6 mmol/L) [21,22].

#### 3.1.1. Differentiation between FCS and MCS

It remains a clinical challenge to differentiate between FCS and MCS, due to the overlap in clinical and biochemical phenotypes. Whilst genetic analysis and post-heparin LPL activity remain the gold standard to differentiate between FCS and MCS, it is also possible to differentiate the two based on lipoprotein electrophoresis or ultracentrifugation, where VLDL and chylomicrons are present in MCS, in contrast to only chylomicrons in FCS [23]. Clinical and laboratory parameters that may help distinguish between FCS and MCS are summarised in Table 2 [23,24].

The difficulty in differentiating between FCS and MCS often results in the diagnosis of FCS being delayed. Some of the rapid biochemical tests to help differentiate between FCS and MCS, e.g., post-heparin LPL activity or lipoprotein electrophoresis, are also not readily available in clinical practice and measuring LPL activity is technically challenging. Moulin et al. proposed an eight-item scoring system to predict the likelihood of FCS [18]. The ability to identify FCS was tested in two independent cohorts using a cut-off score of 10 and yielded a sensitivity and specificity of 88% and 85%, respectively (Figure 2). It is interesting to note that in this score, body mass index (BMI) was not included despite data suggesting a greater prevalence of overweight and obesity in MCS as compared to FCS [23,25]. In a comparative analysis of an FCS and MCS cohort from two phase III clinical trials, volanesorsen and triglyceride levels in familial chylomicronaemia syndrome (APPROACH) and the efficacy and safety of volanesorsen in patients with multifactorial chylomicronaemia (COMPASS), BMI and a history of AP were reported to be the strongest predictors of FCS which, when used in conjunction with one of fasting LDL-C or apolipoprotein B100 (ApoB100), provides a sensitivity and specificity of above 90% [25]. A higher chylomicron-TG-to VLDL-TG-ratio (CM-TG/VLDL-TG) and TG-to-ApoB100 ratio (TG/ApoB100) were also found in the FCS cohort and, in fact, a CM-TG/VLDL value of 3.8 was found to be 100% sensitive and a value of 4.5 to be 100% specific in diagnosing FCS [26]. The addition of these biochemical markers to the eight items of the clinical score may enable a clinician to predict the likelihood of FCS and offer genetic tests to the appropriate cohort, particularly in resource-restricted settings.

#### 3.1.2. Genetic Basis of Familial Chylomicronaemia Syndrome (FCS)

FCS is characterised by a marked reduction in LPL activity with a resultant accumulation of chylomicron particles and SHTG. The most common cause of FCS is pathogenic variants of the *LPL* gene, which is located on the p22 region of chromosome 9 [27]. In a study of genetically proven FCS patients in a phase III clinical trial, 82% had biallelic loss-of-function (LoF) variants in *LPL.* More than 114 variants including nonsense, missense, or frameshift variants have been described [28,29]. Although biallelic LoF variants of *LPL* account for most cases of FCS, LoF variants in other genes encoding proteins responsible for maturation, stabilisation, transport, anchoring, and activation of LPL on endothelial surfaces account for rare forms of FCS. Of the 11 patients from the APPROACH trial who had non-LPL variants, biallelic variants in *GPIHBPI1* were found in 5 (45%), *ApoA5* in 2 (22%), and *LMF1* and *ApoC2* in 1 (11%) each. In addition to this, rare cases of variants of *GPD1* and *CREB3L3* alleles have been reported (Table 3) [28,30,31]. In addition to biallelic homozygous LoF variants, compound heterozygous (a different variant on each allele of one gene locus) and double heterozygous (pathogenic variants on two different gene loci) have also been described in a small subset of patients. 

Data on the natural history and long-term outcomes based on the genetic makeup of FCS are limited. Hegele et al. used the dataset of the phase III clinical trial of volanesorsen and compared the baseline phenotype of LPL FCS and non-LPL FCS. They observed similarities in most of the traits including the prevalence of AP, TG concentrations, BMI, and other lipoprotein concentrations; however, the non-LPL FCS cohort displayed higher post-heparin LPL activity and postprandial insulin and C-peptide levels [28]. The greater tendency of insulin resistance in non-LPL FCS may suggest a contribution of this secondary factor to severe hypertriglyceridaemia or the effect of non-LPL gene products in governing insulin sensitivity. In a longitudinal analysis of LPL FCS (*n* = 10) and non-LPL FCS (*n* = 2), over a median period of 44 months, D’Erasmo et al. observed relatively better control of hypertriglyceridaemia in non-LPL FCS, albeit the prevalence of AP and recurrence were similar [17].

### 3.2. Familial Dysbetalipoproteinaemia

Familial dysbetalipoproteinaemia (FDBL) is a consequence of *ApoE2/E2* homozygosity at the polymorphic *ApoE* gene locus on chromosome 9. *ApoE* has three isoforms, E2, E3, and E4, present on the surface of TRL and is responsible for hepatic uptake of these remnant particles via LDL, LRP1, or heparan sulphate proteoglycan (syndecan 1) receptors. While *ApoE3* is the most common isoform, *ApoE2* homozygosity reduces the binding affinity of TRL to target receptors and, hence, reduces the clearance of remnant particles. This along with the presence of secondary factors lead to enhanced VLDL production and resultant severe hypertriglyceridaemia. The interaction of acquired and genetic factors is required for phenotypic expression [32]. In addition to homozygous *ApoE2* variants, in a minority of individuals, it is caused by an autosomal dominant pathogenic variant in *ApoE*, rather than the more typical *ApoE2/E2* genotype [33]. Whilst palmer crease xanthomas are pathognomonic for FDBL and are present in 18–20% of affected individuals [34,35], eruptive xanthomas, tendon xanthomas, and xanthelasmas are also seen as in other familial lipid disorders. Delayed clearance, increased circulatory time, and preferential entrapment of remnant particles and VLDL pose individuals with FDBL at a higher risk of incident and subsequent ASCVD events [36]. Clinically, it remains a diagnostic challenge to differentiate FDBL from other forms of mixed dyslipidaemias. Beta quantification following ultracentrifugation or gel electrophoresis pre-stained for lipoproteins might help to differentiate, but these methods are technically demanding and not available in routine clinical practice. The measurement of ApoB100, ApoB100-to-total cholesterol ratio (ApoB100/TC) ratio, and VLDL-cholesterol-to-TG ratio (VLDL-C/TG) can be of diagnostic utility. Regardless, the presence of remnant particles and VLDL and their increased circulatory time increase ASCVD risk [33,37].

### 3.3. Lipodystrophies

Lipodystrophies are a complex heterogeneous group of disorders characterized by partial or generalized loss of adipose tissue that can be familial or acquired. A reduced capacity of adipose tissue to expand to enable the storage of fat and a reduced capacity to buffer postprandial lipaemia leads to lipotoxicity, increased oxidative stress, mitochondrial damage, endothelial dysfunction, ectopic fat distribution, and a host of metabolic disorders, i.e., insulin resistance, hepatic steatosis, hypertriglyceridaemia, pancreatitis, and cardiomyopathy [38] Congenital generalized lipodystrophy is an autosomal recessive condition characterized by the near-total generalized absence of body fat. Severe hypertriglyceridaemia ensues because of delayed catabolism of TRL partly due to severe insulin resistance and, hence, the dampened effect of LPL [39]. Recurrent pancreatitis is often more common in individuals with suboptimal glycaemic control [40]. Familial partial lipodystrophies (FPLD) are rare autosomal dominant inherited disorders with variable phenotypic presentation characterized by selective loss of adipose tissue from extremities and the trunk. The two most common types of FPLD are the Dunnigan (type 2; mutation in the *LMNA* gene), and Köbberling (type 1; no mutations in single genes have been identified so far, and polygenic background has been described) varieties. In a comparative analysis of 13 subjects of the Köbberling phenotype with controls, Herbest et al. demonstrated an adverse metabolic phenotype in the Köbberling group which had a mean ± SD TG of 43 ± 12.2 mmol/L [41]. Similar observations were recorded in other comparative studies where Dunnigan and Köbberling phenotypes were found to have an adverse metabolic profile with raised TG, insulin resistance, impaired glucose tolerance, and cardiovascular complications [42,43]. Lazarte et al. have recently demonstrated that in FPLD, the prevalence of severe hypertriglyceridaemia and the tendency to develop AP are greater with the concomitant presence of diabetes [44]. In addition to the familial form, acquired generalized, partial, and localized lipodystrophies have also been described as associated with hypertriglyceridaemia [45] 

## 4. Secondary Hypertriglyceridaemia

Certain metabolic conditions and drugs are often, but not always, associated with varying degrees of severity of hypertriglyceridaemia. This implies that some genetic polymorphisms make an individual more prone to developing variable degrees of hypertriglyceridaemia. Table 4 summarizes the common causes of acquired hypertriglyceridaemia [46,47].

## 5. Complications of Severe Hypertriglyceridaemia

### 5.1. Acute Pancreatitis

AP is an acute inflammatory process involving the pancreas characterized by abdominal pain and the elevation of pancreatic amylase and lipase. Hypertriglyceridaemia-induced pancreatitis (HTGP) is one of the common causes along with gallstone and alcohol-induced pancreatitis and constitutes 5–15% of total cases of pancreatitis [48,49]. Although the prevalence of HTGP in the general population is low, the prevalence of AP in individuals with severe hypertriglyceridaemia is significantly greater and increases sharply at TG levels above 20 mmol/L to more than 15% [21,22]. Severe hypertriglyceridaemia in acute pancreatitis is often an under-recognized entity as the severity of hypertriglyceridaemia and chylomicronaemia disappears quickly after the first few days of cessation of oral feeding—the cornerstone of the management of acute pancreatitis, regardless of aetiology. Despite the presentation of HTGP being very similar to that of AP of other aetiology, its disease course tends to be more severe, with a higher risk of pulmonary and circulatory failure, more intensive care admissions, a greater likelihood of pancreatic necrosis, and a higher median hospital stay and mortality rate [50,51]. In SHTG of monogenic origin, persistent exposure to chylomicronaemia starting at a younger age can lead to severe disease and worse outcomes. The prevalence of AP, recurrent AP, disease severity, need for intensive care unit (ICU) admission, and mortality are higher in FCS as compared to non-FCS hypertriglyceridaemia (Table 5) [17,23,24,25,52,53,54,55].

The greater prevalence of AP, recurrent AP, increased number of hospital admissions, greater length of hospital stay, and increased ICU admissions pose a significant burden to health care resources. In a large US claim database, health care resource utilisation was found to be greater in individuals with chylomicronaemia for related morbidities. This was particularly significant in individuals who had history of pancreatitis. The total cost incurred for patients with chylomicronaemia was 11% higher than those for age, gender, and metabolic phenotype-matched controls. This was substantially higher, approaching $33,000 per patient per year in the subgroup with pancreatitis as compared to $8000 in the whole cohort of chylomicronaemia (with or without pancreatitis) [56]. We believe that economic burden is substantially higher in FCS as compared to MCS due to higher prevalence of pancreatic complications. In addition to this direct health care cost, there are indirect economic implications of chylomicronaemia syndrome-associated complications leading to reduced work productivity and temporary or permanent work loss due to frequent episodes of decompensation. In a global web-based survey of 166 FCS patients, 40% of patients were reported to be un-employed and over 90% of them believe their employment status was compromised because of FCS. Over half of the previously employed individuals attributed their unemployment to FCS [57]. 

The precise pathophysiology of HTGP is subject to debate and many of the proposed theories have been derived from animal models (Figure 3). The greater preponderance of persistent chylomicronaemia in FCS patients from the early years of life could explain the greater frequency and severity of AP in FCS. Further studies are needed to explore this along with the natural history of AP and its correlation with the genotype of FCS.

### 5.2. Atherosclerotic Cardiovascular Disease

Uncertainty about the involvement of chylomicronaemia in atherosclerosis has stemmed from the fact that chylomicron particles are too large to penetrate the vascular intima and, hence, have traditionally been thought as having no role in atherogenesis. However, in most cases of severe hypertriglyceridaemia, a contribution to the increased TG concentration comes from remnant particles. They not only have access to the subendothelial space but are preferentially trapped in the vascular intima due to their size. They can be taken up by macrophages directly without the need for modification and catabolism by tissue LPL. This leads to the production of free fatty acids (FFA) and glycerol, creating a pro-inflammatory milieu with a subsequent atherogenic effect [60]. Due to the rarity of SHTG, young age of onset, and focus on the greater prevalence of life-threatening AP, the risk of ASCVD have not been examined properly. Recently, a population-based study from England has demonstrated an increased risk of myocardial infarction (MI) in persons with severe hypertriglyceridaemia in a univariate model, (TG 10–20 mmol/L; HR 1.85 (1.60–2.15), TG > 20 mmol/L HR 2.04 (1.49–2.78)); however, when controlled for other covariates, this risk failed to be statistically significant in severe hypertriglyceridaemia M (TG 10–20 mmol/L; HR 1.03 (0.88–1.19), TG > 20 mmol/L HR 1.02 (0.75–1.40)), suggesting other factors contributing to the increased risk of MI, without a significant contribution from severe hypertriglyceridaemia [12]. In a retrospective longitudinal cohort analysis on three Italian local health units, as was found in the CALIBER study, individuals with TG levels > 5.7 mmol/L were found to have an increased risk of ASCVD (HR 2.30, 95% CI 1.02–5.18); however, the risk of ASCVD when TG levels > 10 mmol/L was not reported [61].

In FCS, the persistence of large chylomicron particles due to the impairment of LPL activity may explain its protective effect on ASCVD (Table 5). However, the difference in age and favourable metabolic profile might explain the relatively low risk of ASCVD in FCS as compared to MCS in cross-sectional studies. Mitropoloue et al. have demonstrated an increased level of factor VII coagulant activity (FVIIc) with hypertriglyceridaemia only in the presence of functional LPL [62]. An increase in FVIIc activity has been implicated as a risk factor in CHD [63], thereby potentially explaining the protective effect of LPL deficiency in FCS. Development of new-onset diabetes due to recurrent pancreatitis [64], insulin resistance as a consequence of hypertriglyceridaemia, and pro-inflammatory milieu [65] may adversely impact cardiovascular risk profile and, therefore, prospective long-term follow-up studies are warranted to investigate the true incidence of ASCVD in this subset of severe hypertriglyceridaemia cohort.

### 5.3. Microvascular Disease

The independent effect of hypertriglyceridaemia on microvascular outcomes is difficult to elucidate and is complex due to the co-existence of other confounders, i.e., diabetes and obesity, that not only are independent risk factors for retinopathy, neuropathy, and nephropathy, but also are important causes of hypertriglyceridaemia.

#### 5.3.1. Retinopathy

A 23% increased risk of progression of retinopathy and the development of high-risk proliferative diabetic retinopathy (PDR) with hypertriglyceridaemia was reported in The Early Treatment Diabetic Retinopathy Study (ETDRS) by Davis et al. [66]. Similarly, in a longitudinal assessment, Diabetes Control and Complications Trial (DCCT), it was identified that total cholesterol-to-HDL-C ratio, LDL-C, and TG levels [67] serve as predictors to the development of macular oedema and hard exudates, although this association was lost when the risk of progressive PDR and lipid profiles were adjusted for glycaemic control [68]. However, hypertriglyceridaemia was not found to be associated with incident non-proliferative diabetic retinopathy (NPDR) in other studies [66,69]. The Hoorn study reiterated the fact that retinopathy is a complex multifactorial microvascular complication and has demonstrated a positive association between hypertriglyceridaemia and prevalent retinopathy [70].

#### 5.3.2. Neuropathy

Microstructural nerve damage in distal symmetrical diabetic peripheral neuropathy, autonomic neuropathy, and cardiac autonomic neuropathy (CAN) is associated with hypertriglyceridaemia [71,72,73,74,75]. D’Onofrio et al. have recently used corneal nerve morphology as a marker of small fibre integrity and have demonstrated significant small nerve fibre damage and clinical neuropathy with severe hypertriglyceridaemia (TG > 5.5 mmol/L) in normoglycemic individuals [76]. Intriguingly, in obese patients with [77] or without diabetes [78] undergoing bariatric surgery, corneal nerve regeneration correlated with a reduction in TG but not with improvement in glycaemia, BMI, or other lipid parameters. In pooled TG results from 35 studies, Chi et al. demonstrated a higher prevalence of hypertriglyceridaemia in patients with diabetic neuropathy and a 36% increased risk of developing diabetic neuropathy with hypertriglyceridaemia; however, the severity of hypertriglyceridaemia was not found to be associated with severity of neuropathy [79]. In the European Diabetes Prospective Complications (EURODIAB) study, TG levels were found to be significantly associated with incident neuropathy even after adjustment for HbA_1c_ and diabetes duration (OR 1.35 (1.16–1.57)) [73].

#### 5.3.3. Nephropathy

Deposition of lipid thrombi in dilated glomerular capillaries leads to lipoprotein glomerulopathy (LPG) with severe hypertriglyceridaemia. This commonly affects individuals with FDBL; however, LPG can also affect patients with SHTG secondary to other genetic variants [80,81,82]. Proteinuria associated with LPG tends to respond to very-low-fat diet-induced reductions in serum TG [80]. High-fat and high-sugar diets, which promote hypertriglyceridaemia, have also been associated with an increased risk of developing chronic kidney disease (CKD) [83,84]. The pathogenic mechanisms include inflammation, oxidative stress, direct lipid deposition in mesangial cells, podocyte and tubular epithelial cells, mitochondrial damage, and atherosclerosis [85,86,87] and these pathways are common in the development of other microvascular and macrovascular complications. In animal models, a high-fat diet has been shown to induce glomerular sclerosis, similar to that seen in renal biopsies of people with chronic glomerulonephritis. Solid plug and fat emboli SHTG have been demonstrated in glomerular capillaries that contribute to the pathogenesis of glomerulosclerosis [88,89,90].

## 6. Management

### 6.1. Acute Management of Hypertriglyceridaemia in Acute Pancreatitis

AP is the key clinical consideration in the acute presentation of SHTG. Despite this, there are currently no universally accepted guidelines for the management of HTGP and no established treatment target for TG lowering in the acute setting. The mainstay of treatment is to abort the supply of chylomicrons to the systemic circulation by either rendering the patient nil by mouth or with fat-free enteral or parenteral feed. However, in clinical practice, the severity of pancreatitis often leads to the consideration and institution of other interventions to rapidly reduce TG concentrations, i.e., insulin infusion or blood purification techniques (plasma exchange/lipoprotein apheresis, high-volume haemofiltration, haemoperfusion or double filtration plasma apheresis) at outset. Institution of interventional strategies in severe pancreatitis, in contrast to a more conservative approach in mild to moderate cases, has resulted in a selection bias and has confounded the outcomes of retrospective observational studies comparing clinical outcomes between different modalities. Interventional strategies have been advocated by certain experts in the context of severe pancreatitis to rapidly reverse hypertriglyceridaemia [91,92] and are deemed to improve the prognosis [93]. However, several recent small-scale retrospective studies have produced conflicting results, thereby not only challenging the notion of the utility of rapid TG reduction early but also the efficacy of invasive interventions in reducing TG rapidly [93,94,95,96]. The only randomized clinical trial comparing plasma exchange with insulin infusion in mild hypertriglyceridaemia-induced pancreatitis failed to demonstrate the superiority of either modality in rapidly reducing TG levels or clinical outcomes [97] Similarly, high-volume haemofiltration has produced a significantly greater reduction in TG at 24 h; however, it was not found to yield superior clinical outcomes, that were comparable with insulin infusion despite a greater proportion of insulin infusion recipients having APACHE II scores ≥8 (67% (*n* = 34) vs. 53% (*n* = 32)) [98].

In a recent meta-analysis of 13 studies including 934 patients, blood purification techniques were found superior for TG reduction only at 24 h. There was no difference in systemic inflammation, length of hospital stay (LOS), local complications, and mortality. When blood purification was compared separately with conventional treatment only and insulin treatment added to conventional treatment, no difference in mortality and LOS was observed between either group. The blood purification cohort was associated with a higher rate of local complications after adjustment for the severity of AP [99]. Similar findings were described in another meta-analysis of 15 studies comparing blood purification and insulin treatment where, despite achieving a favourable reduction in TG levels with blood purification, systemic inflammation, organ failure, local complications, LOS, recurrence of AP, and mortality were comparable between blood purification and insulin treatment [100]. It should be noted that in both these meta-analyses, very few studies employed blood purification modalities other than therapeutic plasma exchange (TPE); therefore, comparing TPE with other modalities was unable to produce reliable results. Overall, in HTGP, there is no convincing evidence that the course of acute pancreatitis is altered even if the plasma TG level is lowered to a greater degree. 

The use of heparin has been proposed as an additional tool to reduce TG levels [98]; however, a direct comparison of heparin with other modalities has not been undertaken to date. In a retrospective observational study, the use of heparin anticoagulation with TPE is an independent risk factor for death, when compared to citrate anticoagulation, whilst the degree of TG reduction achieved via TPE with or without heparin was similar [101]. Heparin releases stored LPL from endothelial cells that may result in a transient increase in LPL activity but eventually depletes vascular endothelium from LPL and, thereby, causes rebound hypertriglyceridaemia [102,103,104]. Rebound hypertriglyceridaemia and the risk of pancreatic haemorrhage have discouraged the use of continuous heparin infusion as a treatment modality in HTGP. From available evidence of lack of efficacy and risk of bleeding [105,106,107], the use of a continuous heparin infusion to achieve a faster initial reduction in TG level should be avoided. 

There is no strong evidence to support the use of blood purification and insulin, as they do not shorten LOS nor do they improve morbidity or mortality. Therefore, their routine use is not recommended, and treatment should be individualised [108]. Currently available therapeutic options for HTGP are summarized in Table 6 [22]. Two randomized controlled trials comparing different modalities of treatments are in progress and will provide more definitive high-quality evidence in the coming few years [109,110].

Given the role of oxidative stress in AP, there is some evidence to support the use of antioxidants in AP; however, definitive data to prove its efficacy in severe AP and chronic pancreatitis are lacking [111,112,113]. Based on a review of the existing literature, we propose an algorithm (Figure 4) for the management of SHTG-induced AP.

### 6.2. Diet and Lifestyle

Restricting fat intake is of paramount importance in the day-to-day management of patients prone to severe hypertriglyceridaemia. A very-low-fat diet constituting <15% of the total caloric requirement (10–20 g) per day is recommended [114]. Implementation of a diet rich in medium-chain triglycerides (MCT) reduces the risk of postprandial lipaemia and pancreatitis in primary hypertriglyceridaemia [115,116]. MCT oil is metabolized via chylomicron-independent pathways and is transported to the liver bound to albumin where it is oxidized [117]. Limiting refined carbohydrates, sugary drinks, alcohol, fatty meals, and oil-based and fried food, and the substitution to fat-free dairy products and incorporation of leafy green vegetables, wholegrains flaxseeds, soybeans, tofu, or walnuts to meet essential fatty acid requirements constitute the core of dietary management of severe hypertriglyceridaemia. Purified MCT oil has a smoking point of 160 °C and, therefore, should either be added to food after preparation or used at low temperatures for cooking. Supplementation with fat-soluble vitamins (A, D, E, and K) should be initiated to prevent deficiency and monitored annually or earlier if deficiency is suspected clinically [114]. Patients should have access to a specialist dietician, not only to provide tailored low-fat dietary advice but also to monitor for macro- and micronutrient deficiency. 

Diet and lifestyle modification remains the cornerstone in the management of severe hypertriglyceridaemia. MCS is extremely sensitive to intensive lifestyle modification along with identification, removal, or control of secondary factors (Table 4). The use of drugs precipitating hypertriglyceridaemia should be minimized. SHTG individuals frequently have obesity, insulin resistance, and metabolic syndrome phenotypes. With an increased prevalence of obesity and diabetes, a parallel increase in the risk of SHTG is expected in the future. Dietary modifications along with exercise and limiting alcohol intake remain critical to reduce the risk of SHTG and associated complications. An amount of 30–60 min of aerobic exercise dampens postprandial lipemia, though the exact mechanism is unclear [118]. Weight loss acquired via caloric restriction, pharmacological measures, or bariatric surgery improves hypertriglyceridaemia via multifaceted approaches including improved insulin resistance, glycaemic control, increase in LPL activity, and reduction in ApoC3 levels [119]. Up to 70% TG reduction can be achieved via diet and lifestyle modifications; however, the degree of TG reduction varies and is dependent upon the extent of weight loss, adherence to fat and carbohydrate restriction, and type and intensity of exercise. 

Adherence to a low-fat diet is challenging for patients, and it has significant psychosocial implications and impacts social interactions, activities with peers and colleagues, as well as family events that may culminate into depression, the feeling of loneliness, and eating disorders [117,120]. In a global web-based survey of 166 patients with SHTG, 93% of patients found it difficult to comply with a low-fat diet and reported symptoms of fear, anxiety, guilt, and helplessness when they exceed the daily recommended dietary allowance of fat intake. This was compounded by the recurrence of symptoms despite adhering to a low-fat diet [57].

### 6.3. Pharmacotherapy

Traditional lipid-lowering medications usually have minimal effect in very severe hypertriglyceridaemia. Randomised controlled clinical trials evaluating the efficacy of conventional therapeutic agents only involved patients with mild to moderate hypertriglyceridaemia and evaluated cardiovascular disease as a primary outcome [121,122,123,124].

The classes of drugs that preferentially target serum TGs are fibrates, omega-3 fatty acids, orlistat, and niacin, which reduce TG levels by 25–50% [125,126,127,128,129] in mild to moderate hypertriglyceridaemia. However, these agents are less effective in severe hypertriglyceridaemia [15]. The major classes of drugs and agents and their effect on TG lowering are summarized in Table 7.

#### Novel Agents

Volanesorsen is the only pharmacological agent that has been approved by the European Medicines Agency (EMA) and The National Institute for Health and Care Excellence (NICE) for use in SHTG of monogenic origin. It is an antisense oligonucleotide inhibitor of ApoC3 mRNA resulting in a reduced inhibitory effect of ApoC3 on LPL (Figure 1). APPROACH was a Phase III, double-blind randomized controlled trial, where volanesorsen reduced serum TG levels in the FCS cohort by 76.5% compared to a placebo. In an exploratory analysis, during the trial period, the rate of AP in patients with a high risk of recurrence was found to be lower in volanesorsen recipients (four vs. zero) [130]. Similar results were obtained in the COMPASS trial where 94% of patients had MCS and 6% had FCS. Results of the COMPASS trial were comparable with those of the APPROACH trial, where mean TG, chylomicron, and VLDL reduction were >70%, along with a reduction in the number of AP episodes in volanesorsen recipients compared to a placebo (five vs. zero) [131]. Thrombocytopenia remains the predominant concern, which appears to be a class effect [132] necessitating close monitoring. Pancytopenia is also a recognized complication of FCS due to hypersplenism [133]. A galactosamine-conjugated apolipoprotein C-III antisense oligonucleotide analogue (olezarcen) is currently under investigation in a Phase III trial that targets ApoC3 mRNA in hepatocytes and has shown similar TG reduction at lower doses, therefore mitigating drug-specific side effects, and the thrombocytopenia experienced with volanesorsen [134]. Several other pharmacological agents that may be used to target high triglyceride concentrations are at various stages of development and are summarised in Table 8 [130,134,135,136,137,138,139,140,141,142,143,144,145]

### 6.4. Bariatric Surgery

As most patients with hypertriglyceridaemia have a polygenic disease, compounded by secondary factors including obesity, insulin resistance, and diabetes, bariatric surgery represents an attractive treatment option with high efficacy in both reversing the secondary factors and reducing the risk of associated complications. In overweight and obese individuals, although the role of bariatric surgery has not been formally evaluated, available data support a reduction in TG and TRL production and enhanced clearance of TRL particles [146]. There are case reports of bariatric surgery in improving lipid profile in FCS patients either by inducing fat malabsorption or improving insulin resistance [147] or reducing recurrent pancreatitis by ameliorating portal hypertension with splenectomy with no effect on serum TG levels [148]. However, these cases should be regarded as anecdotal in nature. Nevertheless, in patients with severe obesity and co-existing metabolic and cardiovascular co-morbidities, bariatric surgery should be considered as a treatment option. 

## 7. Conclusions

SHTG and CS are rare but represent a major clinical burden with significant implications regarding health care resources. Acute pancreatitis is the most feared complication of SHTG and is associated with higher mortality and morbidity as compared to other causes of AP. Most individuals with SHTG are polygenic, wherein the presence of a secondary factor, e.g., diabetes, obesity, alcohol, or drugs, make them more susceptible to developing SHTG. FCS constitutes a minority of SHTG and is associated with increased risk of pancreatic complications and lower risk of cardiovascular complications compared to MCS. The prevalence of AP, recurrent pancreatitis, hospital admissions, length of stay at hospital, and ICU admissions are greater in FCS as compared to MCS. The cost of health care utilization in individuals with chylomicronaemia is significantly higher than that of normo-lipidemic individuals and is substantially higher in individuals with acute pancreatitis. 

Low-fat diets and the elimination of secondary factors remain cornerstones in the management of SHTG. We propose a possible algorithm to approach a patient with SHTG (Figure 5). SHTG of monogenic origin is poorly responsive to conventional lipid-lowering therapy. Volanesorsen is the only pharmacological agent that has been approved by EMA and NICE for use in FCS. Several therapeutic agents in the experimental phases of use may be of utility in CS in future.

There are no standardised management guidelines for HTGP; however, based on available evidence, we propose an algorithm for the management of HTGP (Figure 4) and advise against the use of insulin in absence of hyperglycaemia and recommend adapting individualized approaches for blood purification techniques.

The phenotype of MCS is quite heterogenous. Further studies are needed to investigate the predictive and prognostic markers for pancreatic and cardiovascular complications in this cohort. Whilst the predominant focus has been to reduce the risk of pancreatitis, the risk of ASCVD with SHTG and chylomicronaemia has not been studied prospectively. The less atherogenic effect of large chylomicron particles, compared to smaller ApoB100-containing lipoproteins, may be offset by recurrent pancreatitis, leading to diabetes, insulin resistance, and SHTG-induced systemic and vascular inflammation. This avenue is yet to be explored. Though the phenotypes of LPL FCS and non-LPL FCS are comparable, comparisons between non-LPL FCS subtypes and genotype–phenotype correlation need to be investigated. Establishing national registries and international collaborative projects are needed to collate sufficient numbers of individuals with this rare pathology to identify and address the unmet needs of this cohort.

## Figures and Tables

**Figure 2 metabolites-13-00621-f002:**
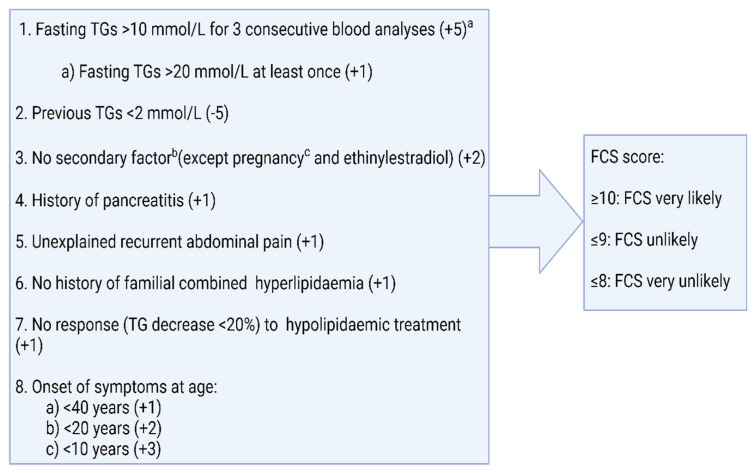
Scoring of familial chylomicronaemia syndrome, according to Moulin et al. [17]. Adapted from: Moulin P., Dufour R., Averna M. et al.: Identification and Diagnosis of patients with familial chylomicronaemia syndrome (FCS): Expert panel recommendations and proposal of an “FCS Score”; Atherosclerosis 2018; DOI: 10.1016/j.atherosclerosis.2018.06.814 [18]. ^a^ Plasma triglyceride concentrations measured at least one month apart. ^b^ Secondary factors include alcohol, diabetes, metabolic syndrome, hypothyroidism, corticotherapy, and additional drugs. ^c^ If diagnosis is made during pregnancy, a second assessment is necessary to confirm diagnosis post-partum.

**Figure 3 metabolites-13-00621-f003:**
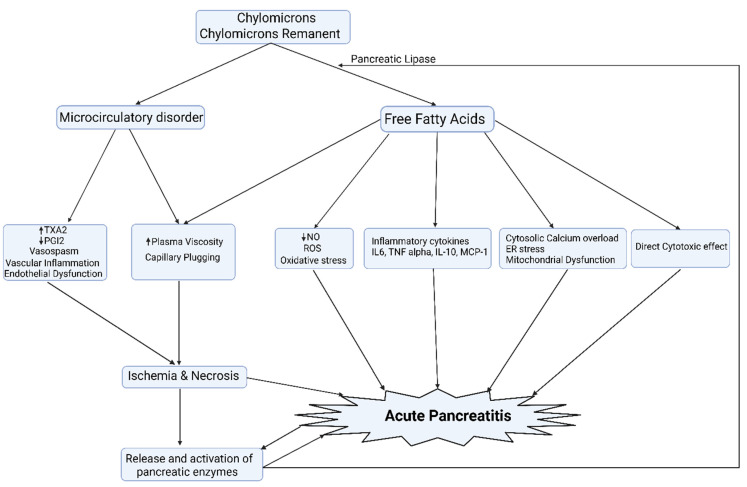
The exocrine pancreas is a rich source of pancreatic lipases that result in the hydrolysis of TG, resulting in the liberation of a large amount of free fatty acids that, once exceeding the binding capacity of albumin, leads to mitochondrial toxicity, oxidative stress, and creates proinflammatory milieu and acinar cell damage, culminating in pancreatic acinar necrosis. In addition, chylomicronaemia and free fatty acids via micelle formation increase the viscosity of blood and, thereby, impede the blood flow to the pancreas, leading to ischemia and worsening acidosis, which potentiates FFA-mediated acinar cell damage. A certain amount of pancreatic lipase, rather than entering the lumen of the gut, finds its way into the capillary circulation of the pancreas. Normally, this does not matter. However, when the pancreatic microcirculation is sluggish in chylomicronaemia, there is time for it to release fatty acids and lysolipids, which are cytotoxic [22,58,59]. ER: endoplasmic reticulum; NO: nitric oxide; PGI2: prostacyclin I2; ROS: reactive oxygen species; TXA2: thromboxane A2.

**Figure 4 metabolites-13-00621-f004:**
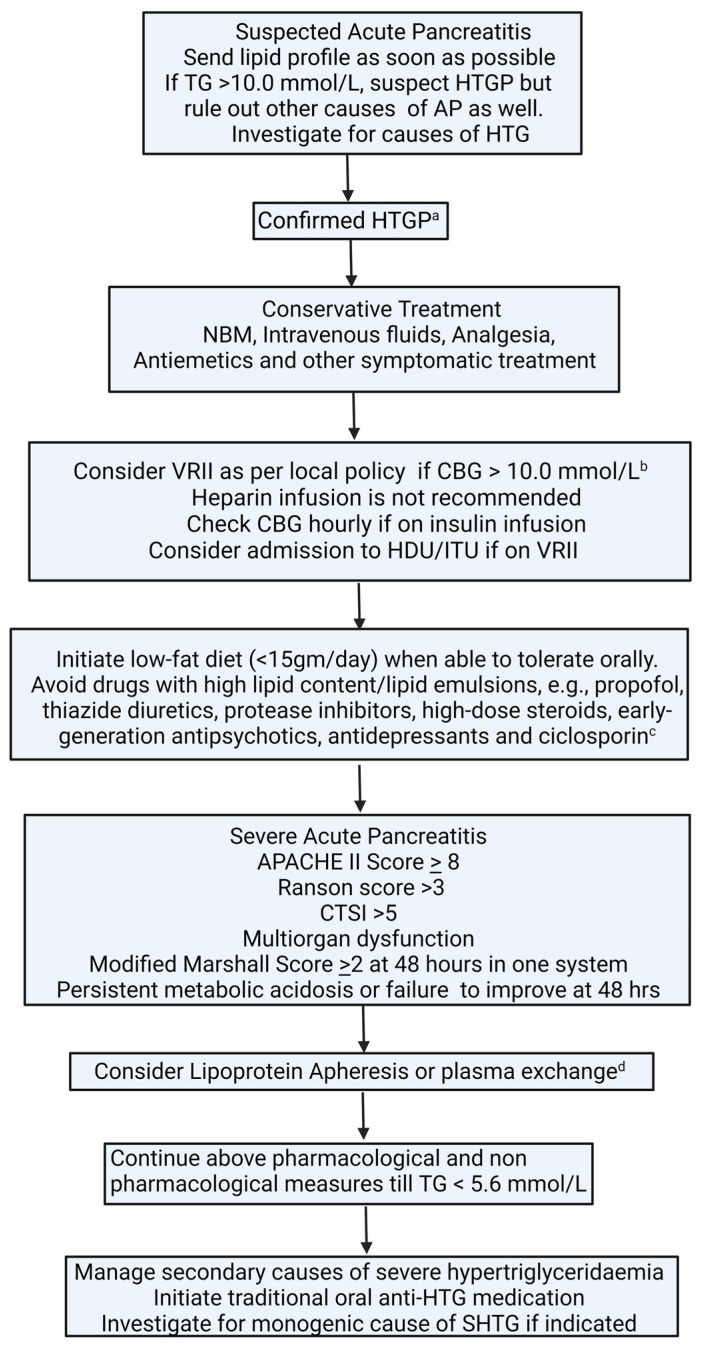
Algorithm for management of hypertriglyceridaemia-induced acute pancreatitis. ^a^ serum amylase can be inaccurate or normal in HTGP due to lipaemia. ^b^ aim to keep CBG between 6.0–10.0 mmol/L. ^c^ seek specialist advice before stopping long-term medications. ^d^ seek a specialist to advise; decision should be individualised. No evidence of benefit in terms of hard clinical outcomes, i.e., mortality, the severity of pancreatitis, length of hospital stay, and complications. AP: acute pancreatitis; APACHE II: The Acute Physiology and Chronic Health Examination; CTSI: CT severity index; HDU: high-dependency unit; HTG: hypertriglyceridaemia; HTGP: hypertriglyceridaemia-induced pancreatitis; ITU: intensive therapy unit; NBM: nil by mouth; SHTG: severe hypertriglyceridaemia; TG, Triglycerides; VRII: variable-rate insulin infusion.

**Figure 5 metabolites-13-00621-f005:**
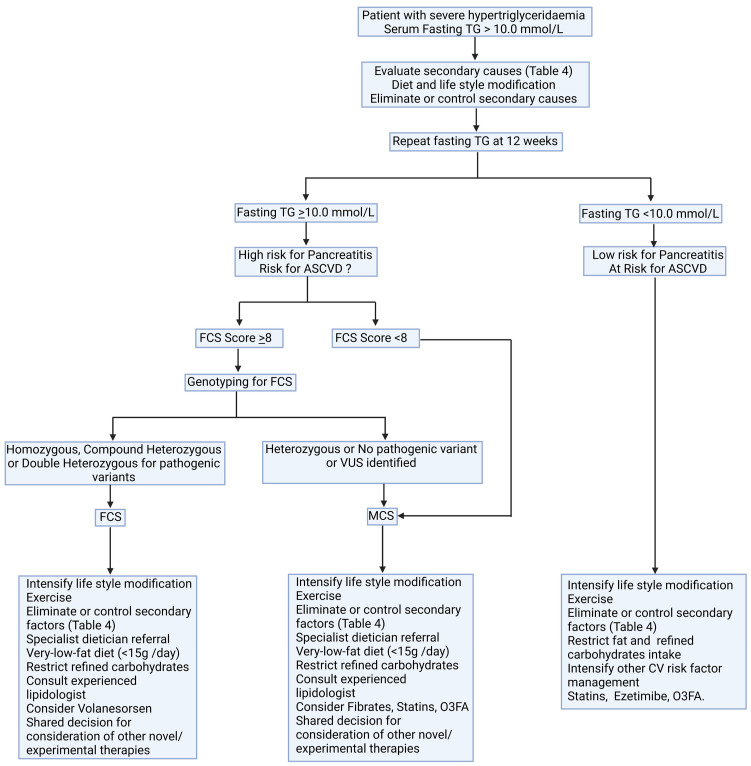
A possible algorithm for diagnosis and management of severe hypertriglyceridaemia. ASCVD: atherosclerotic cardiovascular disease; CV: cardiovascular; FCS: familial chylomicronaemia syndrome; MCS: multifactorial chylomicronaemia syndrome; O3FA: omega-3 fatty acids; VUS: variant of unknown significance.

**Table 2 metabolites-13-00621-t002:** Comparison between FCS and MCS phenotypes.

Features	FCS	MCS
Age of onset	Paediatric or Early Adolescence	Adulthood
Prevalence	1:100,000–1,000,000	1:600
BMI	Normal BMI	Overweight or obese
Contribution of secondary factors	Minor	Major
Peak TG	Higher	Relatively lower
Lowest TG	Higher	Relatively Lower
Acute pancreatitis (prevalence)	Significantly higher	High
Multiple acute pancreatitis (prevalence)	Significantly higher	At the risk of recurrent pancreatitis
Hospital admissions	More frequent	Less frequent
Lipoprotein disturbance	Increased number of chylomicron particles. Reduced LDL and HDL.	Increase in the number of TRL, i.e., VLDL, IDL, VLDL, and chylomicron remnant particles.
ASCVD	Increased?	Significantly higher than FCS
Adverse metabolic phenotype *	Less prevalent	More prevalent
Post-heparin LPL activity	Severely reduced	Normal or mildly impaired
Inheritance pattern	Autosomal recessive	No discrete pattern
Genetic Causes	Presence of two pathogenic variants in *LPL, ApoC2, ApoA5, GPIHBP1, LMF1, GPD 1*, or *CREB3L3*	More than 300 SNVs identified
Response to traditional LLT **	No effect	Mild to moderate effect
**Treatment**	Very-low-fat diet Volanesorsen	Low-fat diet Addressing secondary factors Variable efficacy with pharmacotherapy

* BMI ≥ 27 kg/m^2^, SBP ≥ 130 mmHg, or DBP ≥ 85 mmHg (or treatment for hypertension) and fasting glucose ≥5.6 mmol/L (or diabetes treatment)—the presence of ≥2 of these. ** traditional lipid-lowering drugs, i.e., statins, fibrates, niacin, omega-3 fatty acids. Apo: apolipoprotein; ASCVD: Atherosclerotic cardiovascular disease: BMI: body mass index; CREB3L3: CAMP-responsive element-binding protein 3-like protein 3; FCS: familial chylomicronaemia syndrome; GPD 1: glycerol-3-phosphate dehydrogenase 1; GPIHBP1: glycosylphosphatidylinositol-anchored HDL-binding protein1; HDL: high-density lipoprotein; IDL: Intermediate density lipoproteins; LDL: low-density lipoprotein; LLT: Lipid lowering therapy; LMF1: lipase maturation factor 1; LPL: lipoprotein lipase; MCS: multifactorial chylomicronaemia syndrome; SNV: single nucleotide variant; TG: triglycerides; TRL: triglyceride-rich lipoproteins; VLDL: very-low-density lipoprotein.

**Table 3 metabolites-13-00621-t003:** Genetic basis of familial chylomicronaemia syndrome.

Genes	Inheritance Pattern	Gene Product Function
*LPL*	AR	Hydrolysis of TG, reduction in the size of chylomicrons and VLDL via depleting TG-rich core
*GPIHBPI1*	AR	Binds LPL from subendothelial interstitial space; transports and anchors it to the luminal surface of endothelial cells
*ApoA5*	AR	Activation of LPL
*ApoC2*	AR	Activation of LPL
*LMF 1*	AR	Maturation, stabilisation, and transport of LPL to the capillary endothelial surface
*GPD 1*	AR	The exact mechanism is unclear; overexpression of mutated genes leads to overproduction and secretion of TG.
*CREB3L3*	AR	Functions as a transcription factor for canonical gene expression

AR: autosomal recessive; CREB3L3: CAMP-responsive element-binding protein 3-like protein 3; GPD 1: glycerol-3-phosphate dehydrogenase 1; GPIHBP1: glycosylphosphatidylinositol-anchored HDL-binding protein 1; LMF1: lipase maturation factor 1; LPL: lipoprotein lipase; TG: triglycerides; VLDL: very-low-density lipoprotein.

**Table 4 metabolites-13-00621-t004:** Causes of secondary hypertriglyceridaemia.

Causes	Mechanism
Obesity, IR, and metabolic syndrome phenotype	Increased production of VLDL due to increased flux of FFA from the expanded adipose tissue mass.
Suboptimal diabetes control	Increased VLDL production and reduced chylomicron and VLDL clearance
Alcohol	Increased chylomicron and VLDL production, increased lipolysis-free fatty acid fluxes from adipose tissue to the liver
Pregnancy	Increased chylomicron and VLDL synthesis, reduced HL and LPL activity, relative IR
Chronic renal failure	Downregulation of LPL and LDLR activity
Hypothyroidism	Reduced LPL and LDLR activity
High-Fat and High-GI food	Increased production of chylomicron and VLDL particles
Multiple myeloma	Reduced clearance of TRL particles and reduced function of LPL secondary to paraproteins binding with them
SLE	Reduced LPL activity due to endothelial damage and antibodies against LPL
**Drugs**	
Thiazide diuretics and beta blockers	Reduced LPL activity
Oral Oestrogen, Tamoxifen, Clomiphene	Increased VLDL production
Corticosteroids	Increased VLDL production due to IR
Protease Inhibitors	Increased VLDL production and reduced LPL activity
First- and second-generation antipsychotics and tetracyclic antidepressants	Increased IR and VLDL production and reduced LPL activity
Cyclosporin, sirolimus, and everolimus	Increased ApoC3 levels and inhibited LPL
Isotretinoin	Increased ApoC3 levels
Propofol	Formulated in a 10% oil-in-water lipid emulsion rich in TG and PL and, hence, increased fat delivery

FFA: free fatty acids; GI: glycaemic index; HL: hepatic lipase; IR: insulin resistance; LDLR: low-density lipoprotein receptor; LPL: lipoprotein lipase; PL: phospholipids; SLE: systemic lupus erythematosus; TRL: triglyceride-rich lipoproteins; VLDL: very-low-density lipoprotein.

**Table 5 metabolites-13-00621-t005:** Studies comparing complications of severe hypertriglyceridaemia between FCS and MCS.

Author	Pancreatitis		Recurrent Pancreatitis		CVD		Comments
	FCS % (n)	MCS % (n)	FCS % (n)	MCS % (n)	FCS % (n)	MCS % (n)	
Paquette et al. 2019 [23] (FCS: 25, MCS: 36)	60 (15)	6 (2)	48 (12)	3 (1)	0 (0)	17 (6)	Acute pancreatitis was the presenting feature that led to the ascertainment of CS in 12% of FCS patients and 3% of MCS patients. The FCS cohort was mostly free of metabolic features (55% vs. 6%) while the majority of MCS patients had a combination of two or more adverse metabolic elements (67% vs. 10%)
Iqbal et al. 2020 [24] (FCS: 38, MCS: 40)	76.3 (29)	27.5 (11)	50 (19)	12.5 (5)	15.8 (6)	30 (12)	Peak and trough triglyceride levels were higher in FCS (47.4 (19.8) and 10.2 (7.37)) as compared to MCS (35.7 (22.4) and 5.2 (6.3)). The phenotype of LPL FCS was comparable with non-LPL FCS.
Ariza et al. 2018 [54] (FCS: 26, MCS: 212)	88 (23)	26 (54)	NR	NR	NR	NR	The median number of AP episodes in FCS was 5 (2–12) vs. 1 (1–2) in MCS. Low-fat and/or low-calorie diet led to a significant reduction in TG levels in MCS. However, TG levels remained unchanged in the FCS cohort regardless of the use of conventional LLT and a low-fat diet.
O’Dea et al. 2019 [25] (FCS: 50, MCS: 106)	86 (43)	21.7 (23)	NR	NR	2(1)	9.4 (10)	Baseline data from two phase III trials for volanesorsen (APPROACH and COMPASS). Patients with FCS in the COMPASS trial were excluded from the analysis. BMI and history of pancreatitis along with ApoB100 or ApoA1 had a sensitivity of >90% for the diagnosis of FCS.
Gaudet et al. 2016 * [55] (FCS: 251, MCS: 1981)	67 (168)	14 (277)	50 (125)	NR	NR	NR	33.3% of AP secondary to FCS required ICU care as compared to 3.4% in non-FCS. Pancreatitis-related mortality was higher in FCS as compared to non-FCS AP (6% vs. 0.55%).
Paquette et al. 2021 [53] (FCS: 28 MCS: 75 **)	61 (17)	18.7 (14)	46 (13)	10.7 (8)	0	16 (12)	Prevalence of acute pancreatitis and recurrent pancreatitis was higher in variant-positive MCS (41% and 23%) as compared to variant-negative MCS (9% and 6%). The prevalence of CVD in FCS (0%) was lower but comparable between variant-positive and variant-negative MCS (18% vs. 15%).
D’Erasmo et al. 2019 [17] (FCS: 12, MCS: 19)	75 (9)	37 (7)	42 (5)	16 (3)	9.1(1)	15.8 (3)	The estimated overall incidence rate of AP was 42 per 1000 person-years in FCS and 13 per 1000 person-years in MCS. No difference in the phenotype of LPL FCS and non-LPL FCS.
Belhassen et al. 2021 [52] (FCS: 29, MCS: 124)	58.6 (17)	19.4 (24)	55.1 (16)	12.1 (15)	10.3 (3)	25 (31)	Longitudinal observational study with a median follow-up of 10 years. Ischemic CVD events in FCS were lower in FCS as compared to MCS but were comparable with controls.
Pooled Results	321/459 (70%)	412/2593 (16%)	65/132 (49%)	32/294 (11%)	11/182 (6%)	74/400 (18%)	-

* Results from a survey of 21 lipidologists from 9 countries; ** variant-positive (heterozygous) MCS: 22; variant-negative MCS: 53. AP: acute pancreatitis; APPROACH: volanesorsen and triglyceride levels in familial chylomicronaemia syndrome; COMPASS: efficacy and safety of volanesorsen in patients with multifactorial chylomicronaemia; CS: chylomicronaemia syndrome; CVD: cardiovascular disease; FCS: familial chylomicronaemia syndrome; MCS: multifactorial chylomicronaemia syndrome; NR: not reported.

**Table 6 metabolites-13-00621-t006:** Management principles of hypertriglyceridaemia-induced acute pancreatitis.

Intervention	Mechanism of Action	Comments
Bowel rest, NBM, IVF, Analgesia (Standard of Care)	Pancreatic rest maintains blood flow to the pancreas, reduces chylomicrons and VLDL production, and reduces HTG burden.	Severe pancreatitis may require a prolonged period of fasting; consider post-ligament of Treitz, enteral feeding or parenteral feeding. Consider fat-free/low-fat enteral parenteral feed. Avoid the use of oil-based medication, e.g., Propofol.
Insulin Infusion	Activates LPL activity to accelerate chylomicron degradation and lower TG levels.	Continuous insulin infusion. The risk of hypoglycaemia may outweigh any potential benefits in patients without diabetes. Consider if CBG is persistently >10.0 mmol/L.
Heparin Infusion	The initial increase in lipoprotein lipase activity converts TG to FFA.	Risk of rebound hypertriglyceridaemia, worsening of lipotoxicity from FFA, and risk of bleeding in pancreatic bed. Not recommended.
Lipoprotein apheresis/ Plasma exchange	Removes TG and inflammatory cytokines. Provides functional LPL (plasma exchange).	May be considered in SHTG with organ failure, worsening systemic inflammation, or acidosis. However, there is no convincing evidence to support including TPE as one of the standard therapies.

CBG: capillary blood glucose; FFA: free fatty acid; HTG: hypertriglyceridaemia; IVF: intravenous fluids; LPL: lipoprotein lipase; NBM: nil by mouth; TG: triglyceride; VLDL: very-low-density lipoprotein.

**Table 7 metabolites-13-00621-t007:** Therapeutic agents affecting triglyceride levels.

Class	Agent	Mechanism of Action	Dose	TG Reduction	Comments
Fibrates	Fenofibrate	Increases TRL catabolism via activation of PPARα and LPL and inhibition of ApoC3 and hepatic synthesis of VLDL	160–267 mg OD	Up to 50%	No effect on gut-derived chylomicrons. Reduces VLDL particles. Minimal or no effect in FCS patients.
	Gemfibrozil *		600 mg BD		
	Bezafibrate		200 mg TDS		
	Pemafibrate **		0.4 mg OD		
Vitamin B3	Niacin	Inhibits HSL, reduces FFA delivery to the liver, inhibits DAGAT II, reduces VLDL production	2 g daily	15–30%	Less effective when TG is very high. Improves HDL function and reduces Lp(a). May worsen diabetes. Not available in Europe for clinical use.
Omega-3 Fatty acids	O3AEE, EPA+DHA (Omacor/Lovaza)	Inhibits VLDL production, Inhibits ApoC 3, increases chylomicron clearance by activating LPL	4 g daily	20–30%	Less effective in chylomicronaemia of monogenic origin. Epanova is not commercially available. Vazkepa is the only purified EPA derivative without DHA.
	O3CA, EPA+DHA (Epanova)				
	IPE (Vazkepa)				
Gut lipase inhibitor	Orlistat	Gastric and pancreatic lipase inhibitors. Reduces the absorption of fat and chylomicron production	120 mg TDS	30–50%	Phase II clinical trial for FCS is ongoing.
Leptin Analogue	Metreleptin	Leptin receptor activator	Weight- and Gender-dependent	30–35%	Approved by NICE for lipodystrophies.

* Increased risk of rhabdomyolysis when given with statins. ** on 8 April 2022, Kowa Research Institute, Inc., declares the decision not to continue the phase III PROMINENT study, after recommendations of the Data Safety Monitoring Board (DSMB). Based on the review of a planned interim analysis, the DSMB determined that the primary endpoint was unlikely to be met. No safety concerns were raised. DGAT II: diacylglycerol acyltransferase-2; DHA: docosahexaenoic acid; EPA: eicosapentaenoic acid; FCS: familial chylomicronaemia syndrome; HSL: hormone-sensitive lipase; IPE: icosapent-ethyl; LPL: lipoprotein lipase; NICE: National Institute of Health and Care Excellence; O3AEE: omega-3 acid ethyl esters; O3CA: omega-3 carboxylic acid; PPAR: peroxisome proliferator-activated receptors; TRL: triglyceride-rich lipoproteins; VLDL: very-low-density lipoprotein.

**Table 8 metabolites-13-00621-t008:** Novel pharmacotherapeutic agents for severe hypertriglyceridaemia.

Drug	Mechanism of Action	Effect on TG	Phase of Development	Comments
**Approved Pharmacotherapies**				
Volanesorsen (ISIS-ApoCIIIRx)	ASO against hepatic ApoC3	50–70% reduction	Approved by EMA and NICE for use in FCS in 2019.	Thrombocytopenia remains a predominant side effect requiring close monitoring. Not advisable to start if the platelet count is <140 × 10^9^/L.
Lomitapide *	MTP inhibitor	Up to 70% reduction	Approved by EMA in 2013 for HoFH.	Individual case reports of the progression of steatohepatitis to fibrosis occurred after 10 years of treatment.
**Pharmacotherapies in development**				
Olezarcen (AKCEA ApoCIII-LRx)	GalNAc3 conjugated ASO against hepatic ApoC3	70% reduction	First Phase III trial is expected to be completed in 2023.	Targets the ASGPR in hepatocytes with similar efficacy as compared to native ASO with 20–30-fold lower dose, therefore minimizing side effects including thrombocytopenia.
Evinacumab	Monoclonal antibody against ANGPTL3	55% reduction	Phase II trial for SHTG and AP expected to be completed in 2023.	Reduces LDL cholesterol by 47% and was approved by EMA for use in HoFH in 2019.
ARO ApoCIII	siRNA against ApoC3	40–70% reduction	Phase II trial is expected to be completed in 2023.	In phase I, along with TG reduction, a dose-dependent increase in HDL (40–80%) was also observed.
ARO-ANG3	siRNA against ANGPTL3	Up to 66% reduction	Phase II trial is expected to be completed in 2024. -	
STT-5058	Monoclonal antibody against ApoC3	-	Phase I trial was expected to be completed in December 2022—no updates at the time of writing.	
**Others, Suspended therapies**				
Alipogene Tiparvovec (Glybera)	Gene replacement	40–60% reduction initially.	Approved by EMA in 2012 for clinical use but withdrawn from the market owing to poor commercial prospects in 2017.	Sustained gene expression and reduced risk of pancreatitis despite the transient effect on hypertriglyceridaemia.
Vupanorsen (AKCEA-ANGPTL3-LRx)	ASO against ANGPTL3	50–60% reduction	Development halted in 2022 after a review of the Phase 2b (TRANSLATE-TIMI) study.	Data from the Phase 2b trial did not support the clinical development of the drug for CV risk reduction or SHTG. It was also associated with dose-dependent hepatotoxicity.
Pradigastat	DAGT inhibitor	40% reduction	No updates since 2015.	

* Not approved for FCS. ANGPTL 3: angiopoietin-like 3, SHTG: severe hypertriglyceridaemia; AP: acute pancreatitis; ASGPR: asialoglycoprotein receptors; ASO: antisense oligonucleotide; DAG: diacylglycerol transferase; EMA: European Medicine Agency; FCS: familial chylomicronaemia syndrome; GALNAc_3_: N-acetyl galactosamine; HoFH: homozygous familial hypercholesterolemia; MTP: microsomal triglyceride transfer protein; siRNA: small interfering RNA; TRANSLATE-TIMI: targeting ANGPTL3 with an antisense oligonucleotide in adults with dyslipidaemia.

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
