# Peer review of "Severe Hypertriglyceridaemia and Chylomicronaemia Syndrome—Causes, Clinical Presentation, and Therapeutic Options"

_metabolites, 2023, doi:10.3390/metabo13050621_

Round 1

Reviewer 1 Report

The reviewed manuscript entitled ‘Severe hypertriglyceridemia and Chylomicronaemia syndrome – Causes, Effects & Therapeutic options’ written by Bilal Bashir et al. gathers information about genetic, environmental, epidemiological, pathophysiological, and clinical factors associated with severe hypertriglyceridaemia.

1. The article is quite well written and structured; however, the style should be revised to be more academic.

2. The provided figures are rather low quality, and images with higher resolution should be provided to the final version of the manuscript.

Author Response

Point 1: The article is quite well written and structured; however, the style should be revised to be more academic.

Response: Thanks for feedback. We have made some changes and editted the language of few sections, rephrased the conclusion of the review that hopefully have addressed your question.

Point 2: The provided figures are rather low quality, and images with higher resolution should be provided to the final version of the manuscript.

Response: Images with 600dpi resolution have been provided to the journal separately. However, for the purpose of manuscript submission when they are copied to word document or pdf document, they might appear to be of low resolution, but images submitted separately are of high resolution, i.e 600 dpi while the requirement is of (minimum) 300dpi. We hope production team will look after this.

Reviewer 2 Report

The review by Bashir et al, titled "Severe hypertriglyceridemia and Chylomicronaemia syndrome - Causes, Effects & Therapeutic options," aims to examine the etiology of severe hypertriglyceridemia (SHTG) and chylomicronemia, its genetic basis, its effects on pancreatic, cardiovascular and microvascular complications, and current and potential future pharmacotherapies.

The general concept of the manuscript is interesting. However, there are some issues to be addressed and gaps to be filled to improve the article.

Major comments.

-In the introduction, the authors should better explain the concept of SHTG, FCS and MCS, before talking about the purpose and adding something about the differentiation between FCS and MSC. In fact, the purpose itself seems to me unrelated to the introduction, which is very general, as it superficially talks about triglyceride and lipoprotein metabolism and not about the real purpose of the paper.

-In section 6.2, the authors mainly talk about diet, but little about other factors that might have an effect on the management of patients with severe hypertriglyceridemia, such as physical activity; given also the psychosocial implications faced by patients on a low-fat diet, it would be interesting to evaluate this aspect as well.

-The conclusions are really meager and uninformative; the authors should expand on them. I would like the authors to provide more information about the impact of these syndromes (FCS and MSC) on the health service, resources, so as to better support the purpose of the article; In fact, this concept was only mentioned, but not expanded upon (page 25 lines 26-28).

Author Response

Point 1: In the introduction, the authors should better explain the concept of SHTG, FCS and MCS, before talking about the purpose and adding something about the differentiation between FCS and MSC. In fact, the purpose itself seems to me unrelated to the introduction, which is very general, as it superficially talks about triglyceride and lipoprotein metabolism and not about the real purpose of the paper.

Response: We have amended the introduction to incorporate the changes suggested by reviewer.

-In section 6.2, the authors mainly talk about diet, but little about other factors that might have an effect on the management of patients with severe hypertriglyceridemia, such as physical activity; given also the psychosocial implications faced by patients on a low-fat diet, it would be interesting to evaluate this aspect as well.

Response: We have added a short paragraph on utility of exercise and weight loss. As the main stay of treatment for severe hypertriglyceridaemia is i) low fat diet ii) removal of secondary factors, we have focussed on low fat diet and briefly addressed secondary factors. We have kept this section short in the interest of word count. Mechanistic details and management of obesity and diabetes is beyond the scope of this review article. Psychosocial implications faced by patient on low fat diet has been summarized in last paragraph of the same section.

Point 3: The conclusions are really meager and uninformative; the authors should expand on them. I would like the authors to provide more information about the impact of these syndromes (FCS and MSC) on the health service, resources, so as to better support the purpose of the article; In fact, this concept was only mentioned, but not expanded upon (page 25 lines 26-28).

Response: We have rephrased the conclusion. A paragraph on impact of chylomicronaemia syndrome on health care resources and its economic implications has been added in section 5.1.

Reviewer 3 Report

The authors have submitted a well-written and comprehensive narrative review, it is clear to me that the authors are experts in the field of severe hypertriglyceridemia and chylomicronaemia syndromes.

The present narrative review is a relevanf update on the topic of severe hypertriglyceridemia and chylpmicronemia syndrome, starting from its causes and pathophysiological mechanisms and ending up with modern methods of diagnosis and current treatment options.

The manuscript is interesting, particularly in the context of dyslipidemia underdiagnosis and the potential pancreatic and cardiovascular complications of these disorders.

The topic is original and adds to the current knowledge on lipidology (as discussed above). It is clearly written and the conclusions are consistent with the evidence and arguments presented and address the main questions posed. The paper is also clinically relevant and comes up with novel diagnostic algorithms that are instruments useful for clinicians practicing a wide range of specialties.

The manuscript is worth publishing in the journal Metabolites with minimal revisions.

My only suggestion is that the authors come up with a diagnostic algorithm for severe hypertriglyceridemia and chylomicronaemia syndromes and present it as a figure/as figures.

Author Response

Point 1: My only suggestion is that the authors come up with a diagnostic algorithm for severe hypertriglyceridemia and chylomicronaemia syndromes and present it as a figure/as figures.

Response: Many thanks for your comments. We have incorporated a diagnostic and management algorithm in the conclusion section (Figure 5)

Round 2

Reviewer 2 Report

The authors have sufficiently answered my questions